Identification of key microorganisms in facultative stabilization ponds from dairy industries, using metagenomics

Irazoqui Jose M. 1
Eberhardt Maria F. 1
Adjad Maria M. 2
http://orcid.org/0000-0002-1147-4485 Amadio Ariel F. 1 amadio.ariel@inta.gob.ar
1 Instituto de Investigacion de la Cadena Lactea (INTA-CONICET) , Rafaela, Santa Fe , Argentina
2 Estacion Experimental Rafaela (INTA) , Rafaela, Santa Fe , Argentina
Collado Maria Carmen
Electronic publication date: 2022 Mar 14
Publication date: 2022
Volume: 10
Electronic Location ID: e12772
Received 2021 Jul 13; Accepted 2021 Dec 19
Copyright: © 2022 Irazoqui et al.
Copyright year: 2022
Copyright holder: Irazoqui et al.
License: This is an open access article distributed under the terms of the Creative Commons Attribution License, which permits unrestricted use, distribution, reproduction and adaptation in any medium and for any purpose provided that it is properly attributed. For attribution, the original author(s), title, publication source (PeerJ) and either DOI or URL of the article must be cited.
License URL: https://creativecommons.org/licenses/by/4.0/

Keywords: Metagenomics, Facultative stabilization ponds, Metagenomics assembled genomes, Metabolic pathways

Funding: FONARSEC FSAGRO-LACTO003 MINCyT SNDG-DG-F6 MINCyT PNBIO-1131043 INTA PNAIyAV-1130034 INTA Jose M Irazoqui holds a CONICET fellowship Maria F Eberhardt and Ariel F Amadio are CONICET fellows This research work was supported by FONARSEC FSAGRO-LACTO003 MINCyT; SNDG-DG-F6 MINCyT; PNBIO-1131043 INTA; PNAIyAV-1130034 INTA. Jose M Irazoqui holds a CONICET fellowship. Maria F Eberhardt and Ariel F Amadio are CONICET fellows. The funders had no role in study design, data collection and analysis, decision to publish, or preparation of the manuscript.

==============================
Wastewater stabilization ponds are a natural form of wastewater treatment. Their low operation and maintenance costs have made them popular, especially in developing countries. In these systems, effluents are retained for long periods of time, allowing the microbial communities present in the ponds to degrade the organic matter present, using both aerobic and anaerobic processes. Even though these systems are widespread in low income countries, there are no studies about the microorganisms present in them and how they operate. In this study, we analised the microbial communities of two serial full-scale stabilization ponds systems using whole genome shotgun sequencing. First, a taxonomic profiling of the reads was performed, to estimate the microbial diversity. Then, the reads of each system were assembled and binned, allowing the reconstruction of 110 microbial genomes. A functional analysis of the genomes allowed us to find how the main metabolic pathways are carried out, and we propose several organisms that would be key to this kind of environment, since they play an important role in these metabolic pathways. This study represents the first genome-centred approach to understand the metabolic processes in facultative ponds. A better understanding of these microbial communities and how they stabilize the effluents of dairy industries is necessary to improve them and to minimize the environmental impact of dairy industries wastewater.

Introduction

The wastewater generated by the cheese industries is composed mainly by different dilutions of milk (or transformed products, like whey), and washing water containing alkaline and acidic chemicals after the cleaning of bottles, tanks and equipment (tools, pumps; Carvalho, Prazeres & Rivas, 2013). Anaerobic treatment has been suggested as the best alternative, based on the composition of the wastewater (Rajeshwari et al., 2000). However, for many small industries it is impossible to cope with the cost of their implementation. Instead, facultative stabilization ponds are the most common treatment technology in developing countries due to their low operation and maintenance costs (Von Sperling, 2007). Breifly, the process consists in retaining the wastewater in ponds long enough so that the natural organic matter stabilization processes take place, combining anaerobic and facultative processes. Normally, these systems are organized in several serial ponds, where the end of a pond is connected to the next one, and the water flux is determined by the height of the water column, avoiding backflow and mixing.

Several studies have been conducted over the years to characterize and compare the microbial community involved in the anaerobic degradation of organic wastewater (Riviere et al., 2009; Campanaro et al., 2016; Tsapekos et al., 2017; Fontana et al., 2018), but few studies have assessed the microbial diversity in stabilization ponds (McGarvey et al., 2005; McGarvey et al., 2007; Belila et al., 2013; Cydzik-Kwiatkowska & Zielińska, 2016). Belila et al. (2013), characterized a system that receives mainly domestic sewage and McGarvey et al. (2007) studied aerobic and anaerobic reactors receiving dairy farm wastewater. To the best of our knowledge, there have not been any studies of facultative systems using culture-independent approaches.

Culture-independent molecular surveys have revealed that only a small fraction of the phylogenetic diversity of Bacteria and Archaea is represented by cultivated organisms (Hugenholtz, 2002). Sequencing the total DNA allows the reconstruction of the complete genome of individual organisms from a complex mixture of microbes, in a process known as metagenomic binning. One of the strategies is to use the differential coverage of contigs across multiple samples, combined with composition analysis (Albertsen et al., 2013). This genome-centred approach allows a more powerful functional analysis compared to a simple gene-centred functional reconstruction (McMahon, 2015).

In this work we analyzed the microbial community composition in two stabilization ponds systems of dairy industries and reconstructed a high proportion of the population genomes using whole genome shotgun (WGS) sequencing. We report the reconstruction of over 100 metagenome assembled genomes (MAGs) using a binning strategy that allowed us to perform a genome-centred analysis of genes and metabolic pathways. The analysis of those MAGs provides the opportunity to improve the knowledge about the metabolic processes carried out in the environment and to look for key organisms specific to these treatment systems. Knowledge about the community composition and functional potential is key to maintain and improve the efficiency and stability of the process and provides the base for future studies regarding the use of stabilization ponds for waste treatment and the environmental impact of dairy industries wastewater in low income countries.

Materials and Methods

Pond sampling and physicochemical parameters measuring

Samples were collected from six full-scale stabilization ponds belonging to two small dairy industries, named CYC and AUR, located in the center of Santa Fe province in Argentina. CYC has four serial treatment ponds of 35 × 50 m. The first two are 2.5 m in depth, and the next two are 1.5 m. AUR possesses two ponds, the first is 80 × 40 m and 1.5 m in depth and the second 90 × 40 m and 2.5 m in depth. For each pond, three samples were taken from different sampling points, to avoid spatial variation. Each time water was collected from the bottom of the pond to its surface, to sample the complete water column. Later, all three samples were combined into one composite sample per pond. CYC samples were taken in April 2013 and AUR samples in September 2013. Historic temperatures measured by the Sistema de Información y Gestión Agrometeorológica (SIGA, INTA, http://siga2.inta.gov.ar/) indicate that the average temperature in the region showed no significant differences when comparing those months.

Temperature, pH and conductivity were measured in the ponds using Horiba U-50 multiparameter device. The remaining parameters were determined following Standard Methods for the Examination of Water and Wastewater by the facility of Universidad Tecnológica Nacional (Rafaela, Santa Fe, Argentina; Table S1).

Sampling, DNA extraction and sequencing

Samples of 1 l were taken from each pond. For DNA extraction, 20 ml of samples were poured through a cheese cloth and then centrifuged at 1,000 RPM for 10 min. Pellets were suspended 1 ml of Tris-EDTA 50 mM pH 8 and frozen. Then, 60 ul of lysozyme (10 mg/ml) in Tris 250 mM pH 8 was added and incubated until melting, and then 45 min on ice. A solution containing 0.5% SDS, 50 mM Tris pH 7.5, 0.4 M EDTA and 1 mg/ml proteinase K was added (120 ul) and incubated at 50 °C for 1 h, mixing by inversion every 10 min. The suspension was then transferred to tubes containing the bead mix from PowerWater® Isolation kit (MOBIO, Ahmedabad, Gujarat), and vortexed for 5 min. Debris was separated by centrifugation at 13,000 RPM for 1 min and supernatant was passed to a clean tube. Finally, a classic phenol-chloroform extraction and ethanol precipitation was followed (Sambrook & Russell, 2006). DNA quantification was done using a Nanodrop 2,000 spectrophotometer (Thermo, Waltham, MA, USA).

The six samples were sequenced using a Illumina HiSeq 1,500 platform (INDEAR), generating 2 × 150 bp paired-end reads. All reads were trimmed using Trimmomatic (v0.36; Bolger, Lohse & Usadel, 2014), with default parameters, to remove adapters and low quality sequences. The coverage of each sample was estimated using Nonpareil (v2.4r1; Rodriguez-R & Konstantinidis, 2014a).

Assembly and binning

First, a taxonomic assignment of the reads was performed with KRAKEN2 (v2.0.6; Wood, Lu & Langmead, 2019) against the standard database. Then a genome reconstruction strategy was designed as follows. In this kind of system, the water flows from one pond to the next one, carrying both organic matter and microbes, therefore there is an important share of the microbial diversity conserved all along the system. This assumption was confirmed with the taxonomic profiling of each pond. Hence, for the binning process, we combined all the data of each system in two data sets, CYC and AUR, and used a differential coverage approach, because treating the different ponds as samples would give more statistical support to the binning process. First, IDBA_UD (v1.1.2; Peng et al., 2012) was used for the assembly and contig coverage for each sample was calculated using Bowtie2 (v2.2.4; Langmead & Salzberg, 2012) and Samtools (v1.3.1; Li, 2011). Then, contigs over 1,000 bp were binned using MaxBin2 (v2.2.5; Wu, Simmons & Singer, 2016). The quality of the binning was assessed using CheckM (v1.0.11; Parks et al., 2015). The bins with high completeness (≥70%) and high contamination (≥5%) were manually inspected to remove any mis-binned contig, taking into account marker genes, GC content, and differential coverage. Finally, the completeness of the definitive set of MAGs was calculated again with CheckM and the taxonomic classification was obtained using GTDB-Tk (Chaumeil et al., 2020) against the GTDB (Parks et al., 2018). For those MAGs classified as the novel Patescibacteria plyhum, we checked the results using the ANVI’O classifier (Eren et al., 2021).

Once the final set was obtained, we compared the MAGs between system using the average nucleotide identity (ANI, Rodriguez-R & Konstantinidis, 2014b), calculated as by Goris et al. (2007), using fragment size of 1,000 bp and window size of 700 bp. Only results with at least 50% of the fragments aligned were considered.

Genes were predicted in all MAGs, and not binned contigs also, using PRODIGAL (Hyatt et al., 2010) and annotated against KEGG (Kanehisa & Goto, 2000), Merops (Rawlings et al., 2018) and CAZy (Cantarel et al., 2009). Only KEGG modules listed under the “Metabolism” were searched for, and were considered present in a MAG if over 50% of the genes were present, and key genes of the pathway were present (i.e., RuBisCo genes for carbon fixation).

Results

Taxonomic analysis

After quality trimming, around 1% of the WGS reads were dropped (Table S2). The average coverage estimated for our samples was 86%, suggesting that the sequencing effort was sufficiently high. The lowest coverage was observed for AUR2, with 82.56%.

Using KRAKEN2, only an average of 22% of the reads per sample were classified, with CYC1 having the most classified reads (30%). A total of 58 different phyla were found, but 50 never represented more than 1% of any given sample (Fig. 1, Table S3). In CYC samples, Proteobacteria was the dominant phyla, averaging 70% of every sample. On the other hand, in AUR1 pond both Proteobacteria (38%) and Actinobacteria (32%) were the predominant lineages, while in AUR2, Actinobacteria was the most represented phylum (62%). The other well represented phyla in all samples were Firmicutes (between 2% and 12%) and Bacteroidetes (between 2% and 7%). Overall, there were no major differences between ponds from the same system.

Figure 1 Taxonomic profiling of the microbial communities in the six stabilization ponds.

Barplot of the relative amount of reads for the main taxonomic lineages found using KRAKEN2 in the six samples studied, against the standard database. (A) Percentage of reads classified for each sample. (B) Taxonomic profile of each sample. Taxa with less than 1% were not considered.

Assembly and binning

The assembly of the 60 Gbp of whole-genome sequences generated 238,725 contigs with length ≥1,000 bp (Table 1). Also, thousands of contigs of ≥10 kbp and 128 contigs of more than 100 kb were obtained. More than 60% of the reads corresponding to each system were mapped to these contigs, which suggests that the quality of the assembly was good.

Table 1 Sequencing and assembly summary of both composed samples: AUR and CYC.

Sample	Bases (Gbp)	Paired reads
(Millions)	Contigs	Predicted genes	
Total	≥1 kbp	≥10 kb	≥100 kb	
AUR	13.3	43.3	461,955	91,676	3,304	25	764,778	
CYC	31.4	72.6	845,042	147,049	6,051	103	1,369,301	

The binning process generated 264 bins (95 for AUR and 169 for CYC ponds, Table S4). The percentage of binned contigs (≥1,000 bp) was similar for both systems: 93% for CYC and 94.5% for AUR. After the manual inspection, 25 bins were improved. Six chimeric bins were detected and split, based on differential coverage among the samples, forming 12 new bins. Further inspection led to discarding nine of these 12 new bins, as they were of low completeness and high contamination, and the recovery of 3 MAGs (INTA.AUR.1003, INTA.AUR.1005 and INTA.CYC.1002).

We reconstructed 35 high quality MAGs, with more than 90% completion and low contamination (Table 2) and 75 draft MAGs with ≥70% completeness and <20% of contamination. For members of the Patescibacteria, the completeness threshold used was lower (≥50%), since their small genome size often leads to marker losses (Albertsen et al., 2013; Parks et al., 2015).

Table 2 Summary of the binning results for both composed samples: AUR and CYC.

Sample	Total bins	Total MAGs	High quality MAGs	Draft MAGs	Domains	Phyla	Classes	
AUR	95	43	14	29	1	14	18	
CYC	169	67	21	46	2	13	18	
Total	264	110	35	75	2	16	23	

Examining the taxonomic classification of these genomes, they were distributed among 16 phyla, and 23 classes, almost all of them from the Bacteria domain (Figure 2). For most phyla, only one class was found; the most diverse phylum is Patescibacteria, with four different classes. Only one archaeal genome was reconstructed, INTA.CYC.1002, from the CYC dataset.

Figure 2 Taxonomic classification of MAGs.

Taxonomic profiles of the samples obtained based on our binning approach. The classification was obtained using GTDBtk on high quality MAGs (assessed with CheckM). Only contigs over 1,000 bp were considered for binning. (A) Percentage of reads found in high quality MAGs, that were classified with GTDBtk; in low quality bins, that were not classified; and reads not binned. (B) Taxonomic classification of the high quality MAGs obtained. The height of the bar is proportional to the number of reads in each taxon.

The most abundant MAGs from the AUR dataset were classified as Actinobacteria, a phylum for which no MAGs were observed in the CYC dataset. According to KRAKEN2, Actinobacteria represents almost 50% of the reads in AUR samples, but only 3% of CYC reads. On the other hand, the most abundant MAG in CYC was classified as Saccharimonadia, a taxon that only represented 0.13% of AUR reads and 0.21% of CYC reads, according to KRAKEN2.

A total of 38 different taxa were observed among MAGs. From those, 14 were shared between both datasets, 10 were only found in AUR and 14 were exclusive to CYC. In general, all MAGs from these distinctive taxa had low coverage (less than 1% of the sample), except for the aforementioned Actinobacteria and INTA.AUR.003, classified as member of the Verrucomicrobia phylum. This last phylum had five members in the AUR dataset, some with high coverage, but was almost absent in the CYC dataset (1 MAG, with low coverage).

We calculated the ANI between MAGs of each data set, to see if there were any species shared between both systems. Only six pairs had aligned fractions over 50% (Table S5), with ANI values over 98%. Half of these MAGs were classified as members of the Clostridia class; a pair of Chromatiales, a pair of Bacilli and two Spirochaetes complete the MAGs that were shared between pond systems.

Identification of Patescibacteria

We found a total of 16 MAGs that were classified as members of the recently described Patescibacteria phylum (formerly known as Candidate Phyla Radiation or CPR) using GTDBtk. In order to corroborate these results, we used ANVI’O, which classified as CPR all these MAGs, but 2 (INTA.AUR.015 and INTA.AUR.064). These two genomes showed high levels of completeness (85% and 84% respectively) but also high levels of redundancy, hence were classified as “Inconclusive” and required manual inspection and contamination removal. The taxonomic classification showed that the MAGs were distributed across four classes, being Saccharimonadia the most abundant (7 MAGs), followed by ABY1 (5 MAGs).

On average, 33% of the genes of each MAG were annotated using the KEGG database. All MAGs lacked a complete TCA cycle, but almost all of them showed evidence of other central carbohydrate metabolism modules, namely M00002 (the core glycolysis module) and M00003 (oxaloacetate conversion to fructose-6P). The only MAG that lacked these two modules was INTA.CYC.027, the only one classified as Gracilibacteria; however, this was the only MAG that showed enzymes to utilize ribose. Also, only INTA.CYCY.027 had complete modules related to the synthesis of amino acids: threonine (M00018), serine (M00020), and methionine (M00035).

Metabolic analysis of the microbial communities

Over two million genes were predicted between both data sets (Table 1), with CYC having almost double the amount of genes than AUR. Since genes in MAGs represent only 15% of the total amount, all genes were considered for the functional analysis. The annotation was primarily focused on metabolic related genes, such as KEGG metabolic pathways, CAZymes and proteases. The biological processes of interest are summarized in Fig. 3 and Table S6.

Figure 3 Main metabolic pathways in stabilization ponds.

Scheme of the main metabolic pathways present in facultative stabilization ponds. Predicted genes in MAGs were compared agaings the KEEG database. Arrows indicate the dataset where the process was found complete, while coloured dots indicate if a key organism was involved in that process and found in both datasets.

In MAGs, on average, 40% of the genes were annotated using one of these databases. On the other hand, only a small fraction (<15%) of genes outside MAGs were annotated, since most of these genes were partial (~80%), which makes it difficult to correctly assign a function to them.

For the hydrolysis of complex molecules, like carbohydrates and proteins, CAZymes and proteases were searched. A total of 21,196 CAZymes were found, from which 38.3% were glycosyl hydrolases (GH), 12.7% carbohydrate esterases (CE), and 1.6% polysaccharide lyases (PL). Every MAG had an average of 28 GH, and 8 CE. On the other hand, a total of 2,787 proteases were found, although the classification of these enzymes is based on the catalytic residues, not substrates nor activity, and does not allow a direct functional analysis.

Several MAGs (63/110) had one or more ways to obtain ammonia from amino acids. For the nitrification stage, the first step of the process (methane/ammonia monooxygenase) was not found in either data set. However, there was evidence of the following steps. On the other hand, a total of 14 MAGs presented the module for dissimilatory nitrate reduction (M00530), while 4 MAGs would be able to denitrify nitrate (M00529), and 4 MAGs have the complete fixation module (M00175).

Most MAGs showed one or more modules related to the acidogenesis stage of the carbon cycle. The acetogenesis stage is divided in two stages in KEGG, the conversion of pyruvate into acetyl-CoA (M00307) and the conversion of acetyl-CoA into acetate that can be catalyzed by five different enzymes. The complete or partial pyruvate oxidation module was found in many MAGs (41), but only seven of them also presented any of the enzymes needed for the next step.

Finally, none of the methanogenesis modules were found complete neither in the MAGs nor among unbinned contigs. However, parts of the pathway for the acetoclastic methanogenesis (M00357), and the hydrogenotrophic methanogenesis (M00567) were found in the unbinned contigs of the CYC data set, and some of the genes from methanol methanogenesis pathway (M00356) were found in INTA.CYC.1002, the only archaeal MAG reconstructed.

Evidence was found for 4 CO2 fixation pathways. In 4 MAGs, the first half of the Calvin cycle (M00166) was found complete, including both RuBisCO subunits (K01601 and K01602). The second half (M00167) was not complete in any of these MAGs, but many of the genes of the module were present (in AUR, four of the seven steps were found; in CYC five steps were found). Interestingly, these 4 MAGs were also the only ones to have a complete module for a photosynthetic system (Anoxygenic photosystem II, M00597). Enzymes for the Arnon-Buchanan cycle (M00173) were found in 33 MAGs, but all of them lacked the ATP-citrate lyase (K15230 and K15230), which is the key gene for this pathway. Some genes for the Wood-Ljungdahl pathway (M00377) and the dicarboxylate-hydroxybutyrate cycle (M00374) were found, but there were many steps missing (at least seven of the total 13) to be considered present.

The three sulfur metabolism modules described in KEGG were found in the MAGs. The full dissimilatory sulfate reduction module (M00596) was present in five MAGs, and partially present in another one. As for the assimilatory sulfate reduction module (M00176), only one MAG had it complete, and other 4 MAGs presented some evidence of it. Finally, in five MAGs was detected the complete SOX complex, related to the oxidation of sulfur compounds, plus another MAG where it was found partially. All these MAGs belonged to Proteobacteria, except for INTA.AUR.004, classified as Bacteroidetes, which only presented the assimilatory reduction pathway.

Finally, genes related to the phosphorus metabolism were searched. We looked for the poly-beta-hydroxybutyrate (PHB) synthesys cassette (Müh et al., 1999), that is formed by three genes, phaA (K00626), phaB (K00023), and phaC (K03821), but can be complemented with an enhancer called phaE (K22881). In the AUR set, 2 MAGs showed the complete cassette, including the enhancer. Meanwhile, in the CYC set, 3 MAGs had the complete cassette, but one of them lacked the phaE gene. This last MAG (INTA.CYC.025), even though it had the three subunits, they were not part of the same contig, hence they were not forming a cassette.

Discussion

In this study, we present the analysis of two facultative stabilization ponds systems. These systems receive the effluents from dairy industries, mainly milk derived by-products, and the microbial communities in these environments are tasked with the degradation of this organic waste, combining both aerobic and anaerobic processes. We based the analysis on a genome-centred approach, attempting to reconstruct the complete genome of these microbes and focusing on complete metabolic pathways, and not just single gene markers.

Firstly, we conducted a taxonomic profiling of the community, using two different approaches. The results obtained classifying the metagenomic reads using KRAKEN2 agreed up to a certain extent with the taxonomic assignment of the reconstructed MAGs. Two of the most abundant MAGs in AUR ponds (INTA.AUR.005 and INTA.AUR.009) were both classified as Actinobacteria and they both showed a significantly higher abundance in the second pond. The same situation was observed in KRAKEN2 results. Since the presence of Actinobacteria in activated sludge was linked to operational problems (Seviour et al., 2008), we can assume that the high abundance of these 2 MAGs could be the related to the high COD and BOD observed in AUR ponds.

In CYC ponds, the most abundant MAG (INTA.CYC.001) was classified by GTDB-Tk as Saccharimonadia (Candidatus Saccharibacteria in the NCBI database, formerly known as TM7), a lineage that was almost absent according to KRAKEN2 (in average, 0.2% of CYC samples). Since KRAKEN2 uses the information available in the NCBI genome database, groups like the recently described Patescibacteria phylum (Parks et al., 2018), which includes the class Saccharimonadia, may be under-estimated because there is an under-representation of complete genomes in the NCBI database.

It is also worth noting that only a small fraction of reads was classified using KRAKEN (13.9–30.8%), therefore the results may be useful to have a raw estimation of variation among the samples of a given set, but strong conclusions should not be drawn from them. We consider that a binning strategy followed by a phylogenomic classification, such as GTDB-Tk, is a more powerful strategy to determine the microbial diversity of an environment (Strous et al., 2012, Sczyrba et al., 2017).

The functional gene annotation was based on KEGG modules, which are a collection of manually defined gene sets in a metabolic pathway (Kanehisa et al., 2021). The two major metabolic cycles present in wastewater ponds are carbon and nitrogen cycle, but the metabolism of sulfur and phosphorus are also interesting in this kind of systems, since sulfate can be used as electron acceptor in anaerobic conditions, and phosphorus compounds, like PHB, are used as energy reservoirs by some microorganisms (Grady et al., 2011). The carbon cycle is divided in two stages: the degradation of small molecules to CO2 or methane, going through different steps: acidogenesis, acetogenesis, and methanogenesis; and carbon fixation, obtaining complex compounds from CO2.

Bacteroidetes, described as Bacteroidota in the GTDB, is one of the most represented classes among CYC MAGs (12), second only to Clostridia (13). These organisms have been reported to be present in different reactors because of their ability to degrade organic compounds into monomers (Sun et al., 2015). Even though they do not show significant differences in complete modules with other MAGs, they do present a large number of CAZymes. The most represented type of CAZymes were glycosyl transferases (GT), with a total of 10,028 putative proteins. It has been reported that the GT2 and GT4 families are the most abundant in bacteria, given their role in the synthesis of the cellular wall (Breton et al., 2006). Glycosyl hydrolases (GH) are the second most abundant type of CAZymes, with 8,126 predicted proteins, and the most diverse class, with 107 different families. Many of the GHs families found, like GH1 and GH2, recognize glucose or galactose moiety, which was expected given that lactose is the main saccharide present in milk, the main source of organic matter in these ponds. A more in depth analysis of GHs was performed in a previous publication (Eberhardt, Irazoqui & Amadio, 2021). Finally, carbohydrate esterases (CE) and polysaccharide lyases (PL) were also found, but the amount of genes found was lower (2,702 and 340 respectively).

The Bacteroidetes MAGs INTA.CYC.034, INTA.CYC.039, INTA.CYC.074, and INTA.CYC.095 have most of the glycosyl hydrolases (GHs) in the whole set, suggesting that they may be better suited to incorporate carbon from the environment (present mainly in the form of lactose). Therefore, Bacteroidetes are probably key members of the communities in facultative ponds (Table 3).

Table 3 Core organisms found and their proposed role in the system.

Taxon	# AUR MAGs	# CYC MAGs	Role	
Bacteroidales	5	12	Polimer degradation	
Clostridia	8	18	Amino acid metabolism, acetogenesis	
Sphaerochaetales	3	3	Acetogenesis	
Chromatiales	2	2	Photosynthesis, carbon fixation, sulfur reduction, denitrification	
Sulfurospirillaceae	1	1	Ammonia reduction	
Desulfovibrionales	2	1	Ammonia reduction, sulfur reduction, nitrogen fixation	
Saccharimonadales	3	4	Unknown	

We observed that almost all MAGs had a way to convert monomers into pyruvate, which is the first step in the carbon cycle. This was expected, since these reactions are part of the glycolysis process, by which most organisms obtain energy. On the other hand, the acetogenesis step was less widespread, with only a few MAGs having the enzymes required to convert pyruvate into acetate. Many MAGs had the complete or partial module for pyruvate oxidation, and some MAGs (13) presented one of the enzymes required to convert acetyl-CoA into acetate, but only seven had both, most of them classified as Clostridia. We also found members of the Spirochaetes class to be present in both samples. These MAGs show the module for the pyruvate oxidation, which agrees with previous reports (Narihiro et al., 2015) that suggest that they play a role in the acidogenic stage of the carbon cycle. Nevertheless, we didn’t find in these MAGs the enzymes needed to convert acetyl-CoA into acetate, which suggests that their role is lesser than other key organisms, like Clostridia. Finally, as Grady et al. (2011) stated, it is generally desirable to design stabilization ponds to produce methane because it is a valuable product. This process has only been described in archaeal organisms (Buan, 2018). The number of archaeal reads in both datasets was low, accounting for less than 1% of the classified reads (except for CYC4, where it was 1.03%), and only one archaeal MAG was reconstructed (INTA.CYC.1002). However, this MAG presented some of the genes of the methanogenesis pathways using methanol (M00356) and methylamine (M00563). Also, most of the genes for the acetoclastic and hydrogenotrophic methanogenesis were found in the unbinned contigs of the CYC dataset. Therefore, we can assume that the microbial community of CYC is able to produce methane, although it is not a crucial step for this kind of environment. It is important noticing that, even though methane is a valuable product, given the design of the ponds, harvesting of methane is an impossible task and most of it would end up released to the environment. Therefore, in a global warming context, the lack of methanogens in AUR system would not be considered a negative trait.

From the six major systems used by autotrophs to fix CO2 (Montoya et al., 2012), genes related to four of them were found. Parts of the Arnon-Buchanan cycle, the Wood-Ljungdahl pathway and the dicarboxylate-hydroxybutyrate cycle were found, but the lack of many genes of the pathways, including key genes like the ATP-citrate lyase, points out that these processes are not carried out by the microorganisms in these ponds. However, the Calvin cycle was almost complete in 4 MAGs, including both RuBisCO subunits. These 4 MAGs (INTA.AUR.070, INTA.AUR,082, INTA.CYC.166, and INTA.CYC.169) were also the only MAGs to present a photosynthetic module and the SOX complex to oxidize thiosulfate. It has been reported that some organisms use thiosulfate as an electron donor during photosynthesis (Falkenby et al., 2011). On top of that, these were the only MAGs capable of producing PHA as a carbon reservoir. All these MAGs were classified as members of the Chromatiales order, and two of them (INTA.AUR.070 and INTA.CYC.166) have an ANI of 99.97%. Given their crucial contribution to the carbon fixation in these environments and their conservation in both pond systems, we consider that these microorganisms play a key role in the system.

The other important metabolic process carried out in this type of environment is the nitrogen cycle, which was not found complete in any of the pond systems. In both cases, the missing enzyme was the methane/ammonia monooxygenase (K10944, K10945, and K10946), which catalyses the first step of nitrification. The ammonia-oxidizing microorganisms fall in a few taxonomic groups (Arp, Chain & Klotz, 2007; Zhou et al., 2015) and, even though these groups were identified in the ponds by KRAKEN2, the number of reads found was really low (around 0.0001%). As for the nitrogen-fixing stage, the complete module was found in four MAGs. Other 6 MAGs had the nifH gene, which is used as a marker gene for nitrogen-fixing microorganisms (Zehr et al., 2003), but the other genes of the module were not found. Hence, we consider that limiting the search only to marker genes could be insufficient, and complete modules or cassettes should be looked for. Two taxa that were predominant according to KRAKEN2 and would also would play a part in the nitrogen cycle are Desulfovibrionales and Sulfurospirillaceae. We were able to reconstruct 3 MAGs of the former (2 in AUR and 1 in CYC), and 2 MAGs of the later (1 in each pond). Members of the Desulfovibrionales family have shown the potential for the dissimilatory reduction of both sulfur and ammonia, so they would be important for the nitrogen cycle and to control the levels of sulfate that may be present in the ponds. Meanwhile, the 2 Sulfurospirillaceae also have presented the module for the dissimilatory reduction of ammonia.

Another taxon that was present in both ponds and could also play a key role in facultative ponds is Clostridiales (8 in AUR dataset and 13 in CYC). Clostridia have been suggested to be able to fermentate amino acids (Militon et al., 2015), and although most MAGs had some modules related to the amino acid metabolism, one of the prevalent taxon with the most modules were Clostridiales, including three of the species that were present in both systems (Table S5). Also, most of the MAGs with the complete acidogenesis stage of the carbon cycle were members of this class. The high amount of carbohydrates and proteins dumped in these ponds would make Clostridiales important members of the community.

We observed that the presence or absence of metabolic pathways in a MAG is related to its genome size: smaller genomes, like Patescibacteria and some Firmicutes (for example, INTA.CYC.065 and INTA.AUR.031), have a limited metabolic potential, having notably fewer genes related to the synthesis of lipids, and nucleic acids (Nelson & Stegen, 2015). Furthermore, as previously reported, we found that all of the reconstructed Patescibacteria lacked the TCA cycle (Wrighton et al., 2012, Castelle et al., 2017), but instead only showed the portion of glycolysis involving 3-carbon compounds, or enzymes related to the metabolism of ribose in the case of INTA.CYC.027, the only Gracilibacteria found. Several MAGs of the Patescibacteria phylum were found in both data sets, and in some cases, represented a big part of the sample (INTA.CYC.001 is the most abundant MAG in CYC1). It has been reported to happen in other environments (Remmas et al., 2017), but their small genomes and the lack of isolates makes it difficult to elaborate a strong hypothesis of their role in the community.

There are other taxa shared between both systems, but that are present in low coverage. We found in both datasets microorganisms that have been previously reported to be involved in the degradation of different organic compounds, like Verrucomicrobia (Cardman et al., 2014), Synergistetes (Militon et al., 2015). and Erysipelotrichales (Seyedi et al., 2020). However, none of those MAGs had complete any of the pathway of interest. Therefore, we consider that the presence of these microorganisms might not be key to the environment, as others previously listed (Table 3).

Conclusions

We were able to carry out a genome-centred analysis of the communities in two wastewater stabilization ponds systems. This approach allowed us to reconstruct 110 MAGs, and study the genes and metabolic pathways present in them. This kind of analysis allows a deeper understanding of the metabolic processes in a given environment, and expands the scope that a single marker gene, either functional or taxonomical, can provide. We found seven key taxa present in both pond systems studied, that would play important roles in the degradation of organic compounds in facultative stabilization ponds. A better understanding of the microbial communities and their metabolic potential is necessary to identify more key actors, and could contribute to the improvement of this kind of systems, so common in developing countries, and minimize their environmental impact.

Supplemental Information

Supplemental Information 1 Summary of physicochemical indicator parameters for each sample.

Click here for additional data file.

Supplemental Information 2 Statistics of the sequencing experiment. Number of reads and bases before and after quality trimming.

Click here for additional data file.

Supplemental Information 3 Absolute number of reads observed for each major taxonomic lineage.

Click here for additional data file.

Supplemental Information 4 Binning results for both composed samples: AUR and CYC.

Genome statistics and taxonomical classification (GTDBtk) for each bin obtained. Coverage for each bin was calculated as the sum of the coverage of each contig in a bin, divided by the number of number of contigs in that bin. To normalize values, each coverage was divided by the number of reads in each dataset over one million.

Click here for additional data file.

Supplemental Information 5 Average nucleotide identity between MAGs.

Click here for additional data file.

Supplemental Information 6 KEGG modules and genes for the different metabolic processes occurring in the environment.

Click here for additional data file.

Supplemental Information 7 GenBank accession for all the MAGs reconstructed.

Click here for additional data file.

Additional Information and Declarations

Competing Interests

Author Contributions

DNA Deposition

Data Availability

The authors declare that they have no competing interests.

Jose M. Irazoqui analyzed the data, prepared figures and/or tables, authored or reviewed drafts of the paper, and approved the final draft.

Maria F. Eberhardt analyzed the data, prepared figures and/or tables, authored or reviewed drafts of the paper, and approved the final draft.

Maria M. Adjad performed the experiments, authored or reviewed drafts of the paper, and approved the final draft.

Ariel F. Amadio conceived and designed the experiments, performed the experiments, prepared figures and/or tables, authored or reviewed drafts of the paper, and approved the final draft.

The following information was supplied regarding the deposition of DNA sequences:

All data is available at GenBank: PRJNA508305, BioSample numbers are available in Table S2 and individual MAGs accession numbers are available in Table S7.

The following information was supplied regarding data availability:

The raw sequence data are available at SRA: SRR8327730, SRR8327731, SRR8327732, SRR8327733, SRR8327728 and SRR8327729.

Raw physico-chemical values are available in Table S1, as were provided by facility of the Universidad Nacional Tecnologica. Since these are not integral to the analysis, we considered them to be adequate as they are presented.

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
