# Peer review of "Identification of key microorganisms in facultative stabilization ponds from dairy industries, using metagenomics"

_PeerJ, doi:10.7717/peerj.12772_

## Round 0.1 · original submission · Major Revisions

Please, revise carefully the comments suggested by reviewers.

Reviewer 1 ·

Basic reporting

This article is clearly written and easy to read. The introduction is concise and provide the necessary elements to follow the experiments and the problematic of the paper. Not being involved in research on this type of system, I was not able to assess a possible lack in the background literature.

Experimental design

The research question is well addressed and worth studying. Experimental is straightforward and performed with advanced technology. In particular, shotgun metagenomics is considered as a very powerful technique. The depth of sequencing is high, which is required for such a complex ecosystem. The methods are well described.

Validity of the findings

Overall, the result are interesting and well supported. Nevertheless, we would like authors to consider the following comments:
.
Here, a first exploration is performed with Kraken, a fast procedure, which relies on the availability of a relevant database. In particular it should contain genomes very closely related to those of the ecosystem studied. It seems that it is not the case here, since it assigned only a minor part of the reads. Kaiju, another fast working software would have been probably better, but would have been still limited by the need of reference genomes closely related to those of the pound. It followed that the results presented in Fig. 1 do not reflect the process well, although it is the only easy-to-read figure that provide a picture of the taxonomical composition of the pounds. It also does not provide the value of the amount of unassigned reads (it could be added as a second “small” bar plot ahead showing the part of Assigned/unassigned reads), which is necessary to estimate the representativity of the bar-plots.
.
The authors used a second approach based on assembly, which provided a much better picture of the ecosystem and the main stream of exploitable data (and this fact is stated by the authors). It would be likely relevant to present the taxonomic profiling of the pounds calculated with the MAGs data in a second figure?
.
Discussion around the CAZymes (lines 296-303). Detailing the availably of enzyme capable of degrading saccharides and related compound is a very important issue in many ecosystems. However, in the case of milk-derived products, the main saccharide present should be residual lactose since milk does not contain other saccharides in significant amounts. What else are the authors thinking?
.
Parts “methane production” (lines315-324). Methane is a desirable product when it is recovered for energy production. In the present case, it is probably not desirable because it is released in the atmosphere and could be considered as greenhouse gas (worse than the carbon dioxide). Thus, the fact that methanogenesis is only a minor pathway here might be considered desirable.
.
Last sentence of Chromatiales section (336-8) “Given their unique physiological characteristics, and its conservation in both pond systems, we can conclude that these microorganisms play a key role in the system”. Based on such a type of argument, it may be hypothesized that each group in the ecosystem plays a key role. For example, Patescibacteria also have unique physiological characteristic and several of their MAGs are present in both sets.
.
Line 364: “We observed that the main driver of gene presence or absence in a MAG is its genome size”. What does this mean? It sounds like a truism. May be authors need to find another way to introduce Patescibacteria;

Additional comments

It would have been interesting to further compare and discuss similarities and differences in taxonomical composition and metabolic pathways between AUR and CYC. Based on the (not very relevant) Kraken analysis they appear very different although the MAG analysis shows the presence of several closely related species. May be the MAG-based taxonomic analysis representation would reveal better similarities.
.
Mention of the method should be added in this graph, in addition of the mention of the % of the reads it represent (see ahead)
.
Legend should be added to each supplementary tables and their list presented.

Reviewer 2 ·

Basic reporting

Writing is very good and conveys the ideas well. Some sections, however, could use a proofread to ensure minor grammar mistakes are corrected. This was more noticeable in the intro section. Table S1 is important to set the stage for the environmental conditions present. Consider placing it in the text if possible. Literature review was sufficient. Tables and figures should have standalone captions. Consider adding addition detail to ensure that a reader has more info to achieve this standalone goal. Consider adding a map, placing the locations of the study sites. This is important for global readership. Supplementary data was good and provided additional information pertinent to the study.

Experimental design

This study presents primary research that will be of interest to researchers involved in understanding the potential for wastewater stabilization improvement for a variety of applications beyond the dairy industry. The goal of the study is clear and will contribute to global knowledge on microbial community dynamics in wastewater stabilization ponds. Work appears to follow standards. Work should be repeatable, except for the sampling protocol. This can be addressed with the addition of further information and justification of procedures used.

Some considerations.
1. The experimental design appears sound. More details would benefit the reader and allow the results to placed in the global setting. For instance, when were the samples taken (time of year, time of day)? I found this in Table S1, but in the main document would be beneficial. Does Santa Fe province experience seasonal influences which might influence the microbial communities present in the ponds? Your samples are spread apart from April to September. If not, your date is simply going to describe the collection. If there are influences, then the date becomes even more important and may explain some of the differences you observed between the two sites. A suggestion of the authors would be to list the date and time of the sample collection. If there is a climatic influence based on time of year for sampling, include a discussion point about the affect this might have on the microbial populations present at the sites.
2. Where were the samples taken at the ponds and how were they taken to ensure they were representative of the pond? Spatial sampling can sometimes show distinct differences within a wastewater stabilization pond environment. For example, were samples taken at different depths throughout the pond and combined into a composite sample. This can be important if there are algal blooms in the pond which will skew results vertically in the water column. A figure showing the waste stabilization ponds and the location of sampling should be included. Suggestion to include more details in your sampling methodology.

Validity of the findings

Findings appear to be valid and supported by supplementary material. Conclusions are linked back to original

Additional comments

It is odd that microalgae were not found in the ponds. One would expect their presences, particularly in ponds which may have operational issues. If you specifically excluded algae, please state this and your rationale.

---

## Round 0.2 · accepted · Accept

The authors have covered all questions and points raised by the reviewers.

Reviewer 1 ·

Basic reporting

adequatly revised

Experimental design

adequatly revised

Validity of the findings

adequatly revised

Additional comments

adequatly revised

Reviewer 2 ·

Basic reporting

Their rebuttal and revisions are acceptable.

Experimental design

Their rebuttal and revisions are acceptable.

Validity of the findings

Their rebuttal and revisions are acceptable.